# New Insights into Pediatric Kidney Transplant Rejection Biomarkers: Tissue, Plasma and Urine MicroRNAs Compared to Protocol Biopsy Histology

**DOI:** 10.3390/ijms25031911

**Published:** 2024-02-05

**Authors:** Andrea Carraro, Piera De Gaspari, Benedetta Antoniello, Diana Marzenta, Emanuele Vianello, Benedetta Bussolati, Stefania Tritta, Federica Collino, Loris Bertoldi, Giuseppe Benvenuto, Luca Vedovelli, Elisa Benetti, Susanna Negrisolo

**Affiliations:** 1Laboratory of Immunopathology and Molecular Biology of the Kidney, Department of Women’s and Children’s Health, University of Padova, 35127 Padua, Italy; 2Laboratory Reference, Euroimmun Italy, 35127 Padua, Italy; 3Pediatric Nephrology, Department of Women’s and Children’s Health, Padua University Hospital, 35128 Padua, Italy; 4Department of Molecular Biotechnology and Health Sciences, University of Turin, 10124 Torino, Italy; 5Department of Clinical Sciences and Community Health, University of Milano, 20126 Milan, Italy; 6Paediatric Nephrology, Dialysis and Transplant Unit, Fondazione IRCCS Cà Granda Ospedale Maggiore Policlinico, 20122 Milan, Italy; 7BMR Genomics srl, 35131 Padua, Italy; 8Unit of Biostatistics, Epidemiology and Public Health, Department of Cardio-Thoraco-Vascular Sciences and Public Health, University of Padova, 35128 Padova, Italy; 9Pediatric Research Institute “IRP Città della Speranza”, 35127 Padua, Italy

**Keywords:** renal transplant, microRNA, biomarkers, subclinical rejection, protocol biopsy, extracellular vesicle

## Abstract

The early identification of a subclinical rejection (SCR) can improve the long-term outcome of the transplanted kidney through intensified immunosuppression. However, the only approved diagnostic method is the protocol biopsy, which remains an invasive method and not without minor and/or major complications. The protocol biopsy is defined as the sampling of allograft tissue at pre-established times even in the absence of an impaired renal function; however, it does not avoid histological damage. Therefore, the discovery of new possible biomarkers useful in the prevention of SCR has gained great interest. Among all the possible candidates, there are microRNAs (miRNAs), which are short, noncoding RNA sequences, that are involved in mediating numerous post-transcriptional pathways. They can be found not only in tissues, but also in different biological fluids, both as free particles and contained in extracellular vesicles (EVs) released by different cell types. In this study, we firstly performed a retrospective miRNA screening analysis on biopsies and serum EV samples of 20 pediatric transplanted patients, followed by a second screening on another 10 pediatric transplanted patients’ urine samples at one year post-transplant. In both cohorts, we divided the patients into two groups: patients with histological SCR and patients without histological SCR at one year post-transplantation. The isolated miRNAs were analyzed in an NGS platform to identify different expressions in the two allograft states. Although no statistical data were found in sera, in the tissue and urinary EVs, we highlighted signatures of miRNAs associated with the histological SCR state.

## 1. Introduction

A major cause of morbidity and mortality in infants and the pediatric population (0–18 years old) is end-stage renal disease (ESRD). Every year, about 5 to 10 children affected by kidney disease progress to ESRD, highly increasing their mortality risk [1]. Kidney transplantation is the treatment of choice for children with chronic kidney disease, providing a better quality of life along with a reduced morbidity and mortality, compared to long-term dialysis [2]. Despite advances in surgical techniques, infection surveillance and effective immunosuppressive therapies that have enhanced the short-term survival of kidney allografts in children, the median kidney allograft long-term survival has remained static to no more than 10–20 years. Indeed, the graft survival rate is about 85% at 10 years post-transplantation, with a progressive decrease to 65% at 20 years post-surgery [3]. Furthermore, after transplantation kidney rejection could also occur. Events such as T-cell-mediated rejection (TCMR), the development of de novo, donor-specific, anti-HLA antibodies (DSA), and the subsequent active antibody-mediated rejection (ABMR) are the major contributors to renal allograft failure in children. The incidence of acute rejection is 10–20%, and early intensive immunosuppression treatment can improve allograft survival [4].

The mechanisms underlying allorecognition and transplant rejection exhibit a level of complexity ruled by the intricate interplay of both adaptive and innate immune responses, which are facilitated by robust inflammatory interactions. The current monitoring approach for transplants relies on traditional indicators of renal function, including serum creatinine, proteinuria, drug blood levels and immunosuppressants. The enduring use of these markers is ascribed to the wealth of accumulated clinical experience, cost effectiveness, and widespread applicability in clinical practice [5]. However, alterations in these markers signal an ongoing rejection process. The contemporary challenge lies in identifying noninvasive markers for subclinical rejection. A subclinical rejection (SCR) is characterized by the presence of histological lesions from mild to severe forms, such as arteritis, peritubular capillaritis, glomerulitis, tubulitis, interstitial fibrosis and tubular atrophy, without clinical alteration of the graft function [6,7]. At present, the SCR incidence rate is between 15 to 50% in transplanted patients. Thus, it is fundamental to identify SCR presence and treat it as early as possible [8]. The early detection of rejection, without alteration of renal function, is achievable through protocol biopsies. Nevertheless, it has to be pointed out that although this method’s results are effective, it does not lack invasiveness and associated risks. Although automated biopsy devices and ultrasound guidance have dramatically reduced the incidence of serious complications, a protocol biopsy could often lead to bleeding or infection [9]. Furthermore, although a kidney biopsy may help to check graft conditions, it certainly does not prevent subclinical lesions.

Therefore, the scientific community has focused its attention on finding less invasive novel biomarkers that are helpful in kidney rejection identification, such as HLA antibodies.

Particularly, the presence of anti-HLA, donor-specific antibodies (DSA) is a crucial point in the development of a humoral graft rejection and seems to lead to the loss of the graft. However, although the immunological HLA dosages are optimal to complete the diagnosis of the histological damage, especially for humoral rejection, the HLA-DSA antibody assay is still neither a prognostic nor a predictive index of kidney damage [10].

In this perspective, the study of microRNAs (miRNAs) has emphasized their potential role to become reliable predictive biomarkers in the nephrological field.

MicroRNAs (miRNAs) are small, single-strand, noncoding RNA molecules of about 20–23 nucleotides that play crucial functions in the regulation of gene expression. These sequences, evolutionarily well preserved, are involved in different biological processes such as development, cell differentiation, apoptosis, fatty acid metabolism and oncogenesis [11]. miRNAs act as post-transcriptional gene expression regulators, modulating the expression of their target mRNAs. They are involved in various biochemical processes, including immune responses and organ transplantation [12,13]. Kidney transplant rejection is a complex immunological process that can be influenced by various molecular factors, and miRNAs have emerged as potential regulators in this context. In addition, physiological and pathological changes can also induce alterations in circulating miRNAs. Thus, numerous studies have investigated the different miRNA signatures as possible diagnostic biomarkers [13,14]. For instance, miRNAs could be used as renal damage biomarkers, helpful in preventing kidney rejection [5,15].

In plasma samples, some independent studies reported the altered expression of circulating miR-142-3p and miR-155 in renal transplant recipients with impaired graft functions due to acute or chronic rejection [16,17]. Other studies found an association between miR-142-3p and the maintenance of tolerance mechanisms in the B cell [18]. More recently, Seo et al. proposed that a three-microRNA acute rejection signature, consisting of hsa-miR-21-5p, hsa-miR-31-5p and hsa-miR-4532, could discriminate recipients with acute rejection from those maintaining a stable graft function. This signature in urinary exosomes has been demonstrated to be a discriminative tool to identify acute rejection in a larger validation cohort [14].

Although these results are highly promising, a gap remains to address the possible use of miRNAs in the pediatric renal transplantation field. Adult and pediatric patients are different in many aspects such as their growth, immune system development, presence of previous infections, dosage and type of immunosuppression. All these characteristics might influence the miRNA expression [4,19].

Although miRNAs are considered good biomarkers, they could be unstable and easily degradable if they are free from ribonucleoprotein or lipoprotein particles. Therefore, the cellular miRNAs are often found included in extracellular vesicles (EVs) to avoid this degradation and ensure their function. The EV lipid membranes protect miRNAs from degradation and remain stable in body fluids [20]. Thus, EVs represent a great source of miRNAs [21]. EVs are bilayer lipid membranes released by all cell types in different biological fluids such as saliva, blood, breast milk, urine and seminal fluid. EVs are divided into different groups based on their size, density, composition and cell origin [22,23]. In the nephrological area, blood/serum and urinary extracellular vesicles could be important to understand what kinds of cells and miRNAs are possibly involved in kidney rejection [24,25,26,27]. In a study on adult kidney-transplanted patients, circulating EVs were analyzed to study the kidney graft function. Three specific microRNAs (miR-21, miR-210 and miR-4639) were associated with chronic allograft dysfunction [28]. To our knowledge, there is no published study related to EVs conducted specifically on pediatric patients who have undergone kidney transplants.

In this study, we conducted a retrospective miRNA expression analysis on 30 kidney pediatric patients transplanted in our center that had a histological report of the protocol biopsy performed at one year post-transplantation. Our population consisted of 15 patients diagnosed with SCR and 15 controls with a normal histology. Based on the availability and quality of the biological specimens present in our laboratory biobanks, the patients were further divided into two different cohorts, as specified in the flowchart (Figure 1). The first cohort included 10 SCRs and 10 controls, all having both a fragment of bioptic tissue stored for RNA analysis and a serum sample collected at the same time of the protocol biopsy. Instead, in the second cohort, we considered five patients with SCR and five controls, who all had a urine sample stored at the same time of the protocol biopsy.

## 2. Results

### 2.1. Population

First cohort: The median age of the 20 pediatric patients transplanted was 11 years (6–16 years) at the time of transplant. The group consisted of 13 female patients and 7 male patients. They underwent the transplant between 2012 and 2015 at our center, and they received therapy with mycophenolate mofetil (MMF) and steroids, in addition to either calcineurin inhibitor: cyclosporine (CsA) or FK506.

Second cohort: The median age of the 10 pediatric patients transplanted was 10 years (5–15 years) at the time of transplant. The group consisted of three female patients and seven male patients. They all received a transplant between 2016 and 2017 at our center, and they were treated the same as the first cohort.

The anamnestic (gender, age, weight, height, naive pathology, creatinine, eGFR, viremia and HLA-DSA) and drug (immunosuppression drugs and supplements) data were analyzed using multivariable statistics. The results show that in our populations, there was no statistical difference between the SCR and normal histology groups; thus, the two groups were homogeneous (data available upon request).

### 2.2. EVS Extraction and Characterization

Both the serum EVs (SEVs) and urinary EVs (UEVs) were characterized as recommended by the Minimal Information for Studies of Extracellular Vesicles (MISEV) 2018 guidelines as summarized in Figure 2. Transmitted electronic microscopy (TEM) evidenced the presence of nanoparticles and nanoparticle tracking analysis (NTA) was used to count from 2.79 × 10^11^ to 9.56 × 10^11^ particles/mL for UEVs and from 4.94 × 10^11^ to 4.1 × 10^12^ particles/mL for SEVs (Appendix A). The EVs isolated from urinary samples had an average diameter of 187 ± 7 nm, and the ones isolated from serum samples, of 121 ± 3 nm. According to the “MISEV” nomenclature recommendation, we can confidently assume to have isolated mainly small extracellular vesicles (sEVs < 200 nm) [23]. These results follow what was declared in the “exosome isolation” kit that we used to purify sEVs from serum samples. A Western blot analysis confirmed the presence of the most common transmembrane markers used to identify sEVs such as CD63 and Flotillin-1, a membrane scaffolding protein essential in the control of exosome cargo sorting [29]. Although there was a certain variability in the tetraspanin expression among the samples, this phenomenon has also already been described in the literature [30,31]. Nevertheless, as shown in Appendix A, samples were positive for at least one of the three markers. The most present was Flotillin-1. As seen from the WB analysis, the EDV sample showed a low marker expression, whereas the EEG sample seems to not show evidence of the three tested markers, probably due to its lower concentration than the other samples (2.79 × 10^11^ particles/mL). Therefore, the apparent absence of marker expression in this sample might be due to the Western blot’s intrinsic lower sensitivity compared to the NTA. It has been already reported that low sample concentrations are not enough to allow for effective epitope detection using Western blot analysis [32]. Indeed, the limit detection for this method is in the high ng/mL range [33].

### 2.3. RNA Quality and Concentration

The qualitative and quantitative analyses were performed using an Agilent 2100 Bioanalyzer (Agilent Technologies, Palo Alto, CA, USA) on tissue samples between 0.7 and 7 ng/µL, with an adequate RNA integrity number (RIN) in a range between 7 and 8.5 [34]. The same evaluation was performed on SEV samples. In this case, miRNA concentrations were found to be low, with a range between 0.06 and 0.52 ng/µL.

The quantitative analysis of total RNA performed for UEVs with the bioanalyzer instrument enabled us to detect low concentrations of microRNA. The quantity of vesicular miRNAs was between 197 and 907 pg/uL. Concerning the serum and urinary vesicles, it was not possible to obtain an RIN, as this type of sample does not have the 18S and 28S ribosomal fractions necessary for the evaluation of the quality of the total RNA [35].

### 2.4. miRNA Profiling

Tissue: The miRNA screening was initially performed on kidney biopsies. The normalization of the read counts produced by miRNA sequencing showed a homogeneous distribution of the samples between the two groups of patients (normal histology and subclinical rejection). The analysis revealed a total expression of 1095 different miRNAs from the biopsies of the patients. A small fraction of this microRNA pool was differently expressed in the two groups. In particular, the statistical analysis showed the overexpression of five miRNAs (miR-142-3p, miR-142-5p, hsa-miR-106b-3p, miR-101-3p and miR-185-5p) in the biopsies with a subclinical rejection compared to those with a normal histology (*p*-value < 0.05%; see Table 1). These miRNAs clustered the patients into two main groups (Figure 3).

Furthermore, the data were stratified using ROC analysis. All miRNAs show a high sensitivity and specificity in patients with a subclinical rejection. Particularly, miR-106b-3p and miR-185-5p seem to be the ones that are better able to differentiate the two groups (AUC area under curve = 0.900 and 0.810, respectively; Figure 4).

sSEVs: The miRNA fraction obtained from serum small extracellular vesicles showed a homogeneous distribution between the two groups. After sequencing, about 100 different miRNAs were observed with different levels of expression; four of the five miRNAs overexpressed in the kidney biopsies with a subclinical rejection were also identified in sSEVs: (miR-142-3p, miR-142-5p, miR-101-3p and miR-185-5p); however, they did not show any statistical significance (*p*-value > 0.05).

sUEVs: The normalization of reads obtained from sequencing has revealed different miRNA expressions between the two groups (SCR/normal histology). The analysis has identified the presence of 522 miRNAs in the sUEVs. Among these miRNAs, a subset including 48 sequences has shown a differential expression in the two analyzed subgroups. For instance, hsa-miR-99a-5p, hsa-miR-155-5p, hsa-miR-514a-3p, hsa-miR-125b-2-3p, hsa-miR-509-3p and hsa-miR-381-3p were overexpressed in SCR, whereas hsa-miR-184-3p, has-532-5p, hsa-miR-187-3p, hsa-miR-542-3p, hsa-miR-99b-3p and hsa-miR-95-3p were overexpressed in the normal histology patient group. These 48 miRNAs are listed in Table 2. The heatmap in Figure 5 shows that the 10 patients clustered into two main groups.

## 3. Discussion

miRNAs are known to regulate immune responses by controlling the expressions of genes involved in immune cell activation, differentiation and function. Certain miRNAs have been implicated in the modulation of T-cell and B-cell responses, which are key components of the immune system involved in transplant rejection too. Different studies have identified specific miRNAs associated with acute and chronic rejection in kidney transplantation. These miRNAs have been observed in the peripheral blood, urine and renal tissue of transplant recipients experiencing rejection, suggesting that miRNAs have the potential to serve as noninvasive biomarkers for kidney transplant rejection [18]. Circulating miRNAs are subject to degradation outside cellular environments; to prevent it, they are encapsulated within extracellular vesicles (EVs). The EVs, made up of a lipid bilayer, protect the miRNAs by acting as molecular shuttles maintaining their key role in the regulation of gene expression, and allowing them to be isolated intact from different intact biological matrices [36].

The identification of a miRNA expression profile has gained a lot of interest, because it could aid in the early detection and monitoring of rejection episodes, allowing for timely intervention. This approach has extreme importance mainly in the pediatric transplantation populations, where life expectancies are high, and the possibilities of returning to dialysis and/or encountering a second transplant must be avoided.

In recent years, many studies have focused their attention on trying to identify new possible prognostic biomarkers useful in identifying the onset of subclinical rejection.

To our knowledge, there are many studies of miRNA profiles in adult renal transplanted patients; however, a clear identification of a useful biomarkers panel has not been defined in the pediatric field yet. Furthermore, no one up to now has investigated miRNA profiles carried by extracellular vesicles in biological fluids in children. Our study aims to identify a miRNA profile useful in predicting subclinical rejection in the protocol biopsies and serum and urinary EV samples of kidney-transplanted children.

In the first cohort of patients, the miRNA sequencing analysis revealed an overexpression of five miRNAs in the biopsies of patient with SCR compared to those with a normal histology. The miRNA overexpressed were miR-142-3p, miR-142-5p, miR-101-3p, miR-185-5p and miR-106b-3p. The first four of these miRNAs were also present in SEVs, but they were not significantly upregulated in the SCR patients. In the literature, miR-142-5p and miR-142-3p have already been reported to be associated with acute renal allograft rejection [37,38]. In addition, a high level of miR-142-3p has been detected in the leucocyte and urinary samples of adult transplanted patients with acute rejection and tubular necrosis [39]. However, in our study, miR-142-3p overexpression did not indicate any of these types of damage. Indeed, in a study by Domenico et al., the adult patients had a clinical rejection, whereas the children had a subclinical condition with mild histological lesions. The miR-101-3p also seems to be associated with rejection outside of the human species. Its overexpression positively correlates with acute kidney injury in patients with multiorgan failure [40]. Recently, circulating mir-101-3p was associated with biopsy-proven chronic allograft nephropathy or rejection in a cohort of adults with ten-year-old transplants [41].

Differently, there are no studies directly linking the expression of miR-185-5p to human allograft rejection. Nevertheless, some recent functional studies have highlighted the involvement of this miRNA in the regulation of TGF-beta signaling, which represents the most important player in renal fibrosis [42,43]. In this perspective, the authors concluded that inactivating TGF beta by miR-185-5p could help avoid graft fibrosis in patients with humoral or cellular rejection. Therefore, its upregulation in the graft could protect against the chronic damage associated with rejection.

The last miRNA significantly overexpressed in tissue samples of rejected patients was miR-106b-3p, which seems to be associated with acute kidney injury. Its function has been recently confirmed by J.M. Hu et al. who elegantly demonstrated that the miR-106b-5p antagonist attenuated ARI in rats and H/R injury in cells [44].

Although we did not find any statistical differences in the miRNA expressions in serum samples, four of the five miRNAs detected in the tissue samples were also overexpressed in the sera. This discrepancy between tissue and serum samples could be due to serum degradation due to storage conditions. Indeed, unlike the tissue samples, no RNAase inhibitors were added to the sera for long-term preservation. Furthermore, in our study, we considered patients having subclinical rejection because we wanted to possibly find early predictive biomarkers to prevent the progress towards a more serious condition. A subclinical rejection is a local phenomenon that does not imply the presence of systemic damage. Therefore, the change in the miRNA expression in serum might be detected only during late phases, when functional and clinical manifestations have already occurred.

In the second cohort of patients, the miRNA sequencing analysis revealed different expressions of 48 miRNAs in the urinary EVs of children with SCR compared to those with a normal histology. The miR-99a-5p, miR-155-5p, miR-514a-3p, miR-125b-2-3p, miR-509-3p and miR-381-3p were the most overexpressed in SCR, whereas miR-184-3p, 532-5p, miR-187-3p, miR-542-3p, miR-99b-3p and miR-95-3p were the most upregulated in the normal histology patient group. All these miRNAs were present in the renal disease literature, confirming their hypothetical function in the positive or negative regulations of kidney damage mechanisms. Among these, the most reliable biomarkers of allograft rejection might be miR-99a-5p, miR-155-5p and miR-125b-2-3p, because they were already found upregulated during clinical rejection in serum, urine or tissue samples, as we found in this study [14,45,46]. Particularly, mir-155-5p was found by Millàn et al. in urinary pellets in association with the proinflammatory chemokine CXCL10 as a prognostic and predictive biomarker of rejection [47], and more recently, in association with immunosuppressive drug exposure as an early prognostic biomarker of acute rejection [48]. Finally, mir155-5p, miR-145-5p and miR-23b-3p were screened as putative biomarkers for KTR monitoring because of an integrative bioinformatics model developed based on multiomics network characterization for miRNA biomarker discovery in KTR; therefore, authors have suggested to investigate these miRNAs through molecular experiments using human samples and to perform further clinical validation [49].

In analyzing the data obtained from this study, it is important to highlight some major limitations: firstly, the small size of the population. The statistical analysis and correlations with some clinical and histological characteristics might lose statistical significance due to the low number. The population analyzed, in total, consisted of 30 pediatric patients at one year after kidney transplant who were enrolled with the same criteria, having similar anamnestic and clinical data and protocol biopsies. This biopsy was performed for the histological monitoring of the kidney in the absence of clinical signs of rejection, such as decreased eGFR, proteinuria and increased donor-specific antibodies. However, the samples available in the laboratory biobanks for these patients were not the same for all. For the transplanted patients enrolled from 2011 to 2016, they were available as a fragment of a frozen protocol biopsy and a serum sample, stored during the biopsy protocol hospitalization time. Instead, patients enrolled from 2016–2017 had a urine sample obtained immediately before performing the protocol biopsy. Therefore, although the patient cohorts are similar, the biological specimens from which the miRNAs were extracted are different in the two study groups. Thus, statistical comparisons are not possible; however, we could speculate on possible similarities or differences in the miRNA identifications in patients with SCR compared to those with a normal histology. Given the uniqueness of the samples analyzed, these findings remain extremely important and should be taken in consideration when trying to identify early biomarkers of kidney transplant rejection in pediatric patients. Furthermore, it is highly encouraging to have identified miRNAs differentially expressed in patients with SCR, which are often found in the literature to be present in the serum and urine of patients having clinical manifestations and bioptic reports of organ rejection.

## 4. Materials and Methods

### 4.1. Patients’ Enrollment and Study Design

In the present study, we enrolled a total of 30 pediatric patients, who were selected from a list of patients who underwent kidney transplantation at the Pediatric Nephrology Unit in the Department of Women’s and Children’s Health, Padua University Hospital. All the patients performed the protocol biopsy at one year after transplantation and were recruited according to the same inclusion criteria reported below.

The inclusion criteria are as follows:Age < 18 years;First transplant;Single transplant (not two grafts, for example, kidney and liver);Not hyperimmune;No surgical complications;No delay in graft function;No clinical or subclinical rejection before one-year post-transplant;Stable graft at one-year post-transplant, with creatinine-based estimates of kidney transplant function or proteinuria.

Two cohorts of pediatric patients with similar characteristics but with different types of samples available were enrolled in our study. The first included 20 patients who were studied for miRNA expression in their protocol biopsies and small EVSs in their sera; the second independent cohort of 10 patients was studied for miRNAs transported by urinary EVSs in correlation with the histology of the protocol biopsies. In both cases, we performed the miRNA sequencing analysis one year after transplant in patients with a stable renal function. Enrolled patients were divided into 2 groups based on the Banff histological classification: no subclinical rejection group (Banff 1 group) had a normal histology, and the subclinical rejection group (classes 2, 3, 4 and 5 from 2017 Banff classification [50]) had histological lesions related to acute or chronic rejection, either antibody-mediated or cellular mediated.

For the first cohort of 20 enrolled patients, a fragment of the needle biopsy (about one tenth) and a serum sample were collected at the same time and stored at −80 °C until their further analysis. The biopsy was treated with RNAlater^®^ (Sigma-Aldrich, Saint Louis, MO, USA) before the freezing procedure. For the second cohort of 10 enrolled patients, a urine sample was collected at the time of the one-year post-transplant protocol biopsy, centrifuged and filtered to remove bacteria and cellular debris (3500 rpm for 5 min at 4 °C and filtered with 0.22 µm filter). Urine samples were stored at −80 °C until the EV isolation. For all patients, we obtained, at the time of the protocol biopsy, informed consent from their parents to store their samples and use their biological specimens for research.

### 4.2. Serum EV Isolation

Small extracellular vesicles were isolated from 250 µL of serum samples using the ExoQuick-^TM^ exosome precipitation solution (System Biosciences, Palo Alto, CA, USA) following the manufacturer’s instruction. The final pellet was suspended in prefiltered PBS (1:10) and used for RNA extraction.

### 4.3. Urinary EV Isolation

Urinary extracellular vesicles (UEVs) were isolated from prefiltered urine by ultracentrifugation, currently considered the gold standard according to the MISEV2018 guidelines (Minimal Information for Studies of Extracellular Vesicles) [23]. Two consecutive spin cycles at 100,000× *g* for 2 h at 4 °C were performed. The first cycle isolated the UEVs from 40 mL of urine; the second one washed the UEVs. The final pellet was resuspended in 600 µL of 1× PBS (prefiltered with 0.22 µm filters).

### 4.4. EV Characterization

To confirm the presence of extracellular vesicles, serum and urine samples were divided into identical aliquots. One aliquot was used for the Western blot analysis, the second for transmission electron microscopy (TEM), the third for quantification using the nanoparticles tracking analysis (NTA) and a fourth aliquot was used for miRNA extraction. The aliquot designated for TEM analysis was supplemented with 1% dimethyl sulfoxide (DMSO), necessary for preserving vesicular membrane integrity.

### 4.5. Western Blot Analysis

The total proteins present in the EVS aliquots were quantified using a BCA test. For protein separation, a 1% SDS polyacrylamide gel was used and transferred to nitrocellulose gels. Membrane strips were incubated with blocking buffer for 1 h at room temperature, and then, primary antibodies were added at the dilution of 1:500 (CD63, CD81 and Floltillin-1 Gene Tex, Alton Pkwy Irvine, CA 92606 USA) overnight at +4 °C. The day after, the blots were washed in TBS with 1% Tween-20^®^ (Sigma-Aldrich, Saint Louis, MO, USA) (TBS-T) three times for 10 min and then incubated with secondary antibodies: the anti-mouse at the dilution of 1:20,000, the anti-rabbit at the dilution of 1:25,000 and the substrate-HRP WESTAR (Cyanagen Srl, Bologna, Italy). Afterwards, the strips were washed again three times for 10 min in TBS-T and then imaged using the I-Bright instrument.

### 4.6. Transmitted Electronic Microscopy (TEM)

The UEV suspension was put onto a glow-discharged, formvar-coated, copper grid 300 mesh (EMS, Hatfield, PA, USA). Excess solution was removed, and the grids were subsequently negatively stained with 1% aurothioglucose (USP) and examined with an electron microscope as described by Collino F et al. [51].

### 4.7. Nanoparticle Tracking Analysis (NTA)

The quantification and evaluation of the vesicular dimensions was conducted using the NTA method with the Nanosight NS3000 instrument (Malvern, UK). All samples were diluted in 1 mL of 1× PBS before being analyzed using the Nanosight NS3000 instrument.

### 4.8. RNA Extraction and NGS Sequencing

The total RNA was isolated from kidney tissue, serum EVs and urinary EVs using the miRNeasy Mini kit (Qiagen, Venlo, The Netherlands) following the manufacturer’s instruction. Tissue samples were treated with QIAzol and mechanically homogenized before the RNA extraction.

The RNA quality and concentration were checked using the Agilent 2100 bioanalyzer. The miRNA sequencing was performed using the NGS sequencing Illumina NextSeq 500 platform with the SMARTer^®^ smRNA-Seq kit (Clontech, Mountain View, CA, USA). As a positive control, we used a known microRNA: miR-163s.

### 4.9. Alignment and Comparison of Sequenced miRNAs

Sequencing data were analyzed using the MiR&moRe2 pipeline (v.0.2.3), which provides the raw count matrix as the final output. All miRNAs with less than 10 counts summed across all samples were excluded from the analysis. Normalization was performed using the DESeq2 library (v3.11) in R software, based on library size, sequencing depth, log2 scaling, VST (variance stabilizing transformations) and dispersion estimation tests. The success of the test was evaluated using boxplots and PCA plots. In the next step, the analysis of differentially expressed miRNAs was also conducted with DESeq2 using the results function (with standard parameters). Once the list of differentially expressed miRNAs was obtained, only those with a *p*-value < 0.05 were selected. The magnitude of differential expression was indicated by logFC, the logarithm of the ratio of a miRNA expression in the two conditions (SCR vs. no SCR). Thus, the genes with a positive logFC are considered upregulated in rejections, whereas a negative logFC indicates upregulation in nonrejections. Additionally, we calculated baseMean, the normalized mean value of that miRNA across the entire sample dataset.

### 4.10. Statistical Analysis

A statistical analysis was performed with the statistical open-source R; in particular, edge-R was considered. The statistical significance was considered with a *p*-value (*p*) < 0.05. The accuracy of the miRNA results was determined using ROC curves and evaluating the sensitivity and specificity parameters.

The anamnestic and drug data were analyzed using multivariable statistics by performing tests for nonparametric data (Mann–Whitney U test for independent data) and categorical data (Pearson’s chi-square test with correction according to Fisher’s exact test). The null hypothesis of a homogeneous distribution of the values observed between the two groups was always accepted (*p* > 0.05).

## 5. Conclusions

The miRNAs isolated from the different serum, urine and renal tissue specimens of our patients showed a different miRNA expression pattern, mostly known in the literature to be associated with the mechanisms of transplanted kidney rejection. In the first cohort, the miRNA sequencing analysis highlighted the significant overexpression of miR-142-3p, miR-142-5p, miR-101-3p, miR-185-5p and miR-106b-3p in the tissue samples of transplanted patients with a histological diagnosis of subclinical rejection versus patients with a normal histology (*p*-value < 0.05%), and four of these miRNAs were detected in serum EVs; in these specimens, there was no statistical difference in the miRNA expression between the two patient populations. This discrepancy between tissue and serum samples could be due to a worse preservation of the sera in our biobank.

In the second cohort, the statistical analysis revealed a signature of 48 miRNAs in the urinary EVs of kidney-transplanted patients with or without histological subclinical rejection. Among these, miR-99a-5p, miR-155-5p and miR-125b-2-3p were the most upregulated in the urinary EVs of patients with a subclinical rejection, and were found upregulated clinical rejection cases in the serum, urine or tissue samples of many studies.

The miRNAs observed to be differently expressed in the biopsies of patients with a subclinical rejection were also observed in the serum and urinary EVs; however, in these sources, they were not statistically associated with subclinical rejection. It could be hypothesized that this is related to their local action in the early stages of subclinical rejection, which is not yet detectable in the circulation or outside the tissue itself.

However, to develop a noninvasive, subclinical rejection diagnostic test, the data obtained from the analysis of urinary vesicles are very stimulating. The study confirms the possibility of isolating EVs even from small quantities of urine and being able to quantify their miRNA content. The data observed highlight the possible association between these miRNAs and subclinical rejection in pediatric transplant patients, and they might be considered as possible early biomarkers useful to prevent rejection and graft loss, but it will be necessary to confirm these results in a larger population.

## Figures and Tables

**Figure 1 ijms-25-01911-f001:**
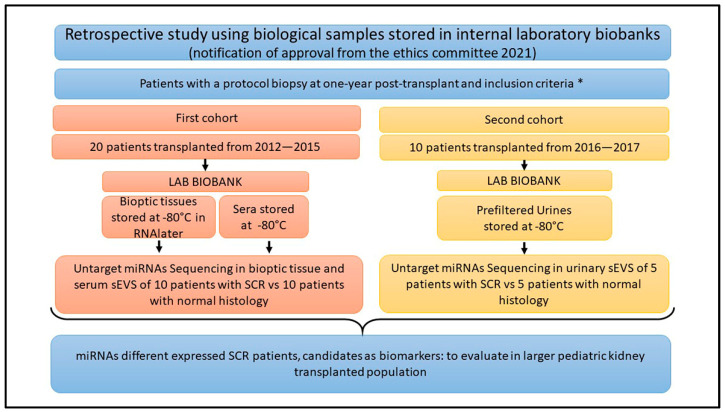
Flowchart of the study design. The figure summarizes the details of the two cohorts analyzed. All the samples were collected one-year post-transplant at the time of the protocol biopsy. The samples were stored at −80 °C in our laboratory biobank after obtaining the informed consent of patient’s parents, for storage and research purpose. Frozen biopsy samples were collected from 2011 to 2015, whereas urine samples were collected starting from 2016. Inclusion criteria * are specified in the Section 4.1. Acronyms: LAB = laboratory; miRNAs = microRNAs; sEVS = small extracellular vesicle; SCR = subclinical rejection.

**Figure 2 ijms-25-01911-f002:**
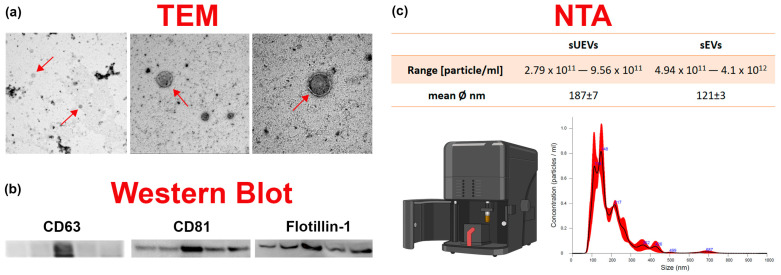
sEV characterization. The image summarizes the tests performed to characterize sEVs in serum (sSEVs) and urine (sUEVs). (**a**) A representative image of sEV identification using transmitted electronic microscopy (TEM); (**b**) a representative image of the main vesicular markers, such as tetraspanins (CD81 and CD63) and Flotillin-1 identified in sEVs using WB analysis. (**c**) The table summarizes the concentration range (particles/mL) of the sEVs obtained and their average diameter. The graph shows an example of a nanoparticle tracking analysis (NTA) output (the Nanosight NS300 image has been created with BioRender.com).

**Figure 3 ijms-25-01911-f003:**
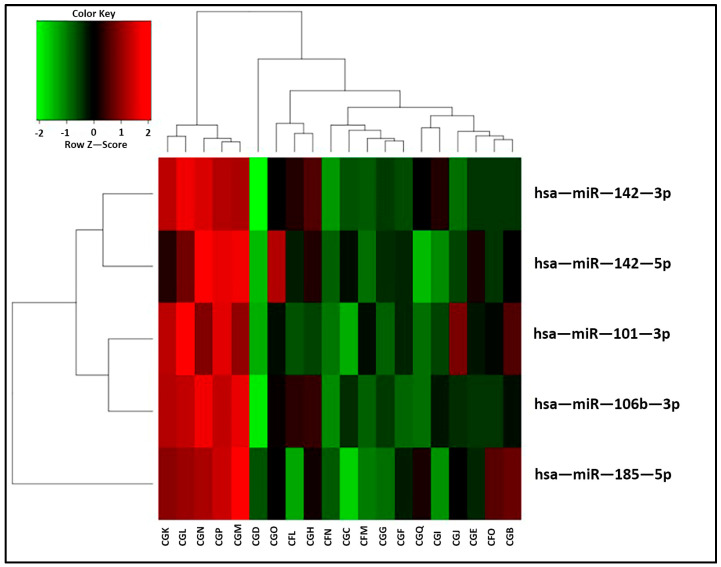
The heatmap represents the miRNAs differentially expressed in the 20 renal biopsies. The red color indicates upregulation, whereas green indicates downregulation. The two groups clustered perfectly. A clear overexpression of the 5 miRNAs (miR-142-3p, miR-142-5p, hsa-miR-106b-3p, miR-101-3p and miR-185-5p) can be observed mainly in 5 out of the 10 samples of patients with subclinical rejection at one year after transplantation (CGK, CGL, CGN, CGP and CGM). (Patients with SCR: CGN, CGP, CGO, CFO; CGK, CGQ, CGL, CGM, CFN and CGJ; patients with normal histology: CGF, CFL, CGI, CGG, CGH, CGE, CGC, CGB, CGD and CFM).

**Figure 4 ijms-25-01911-f004:**
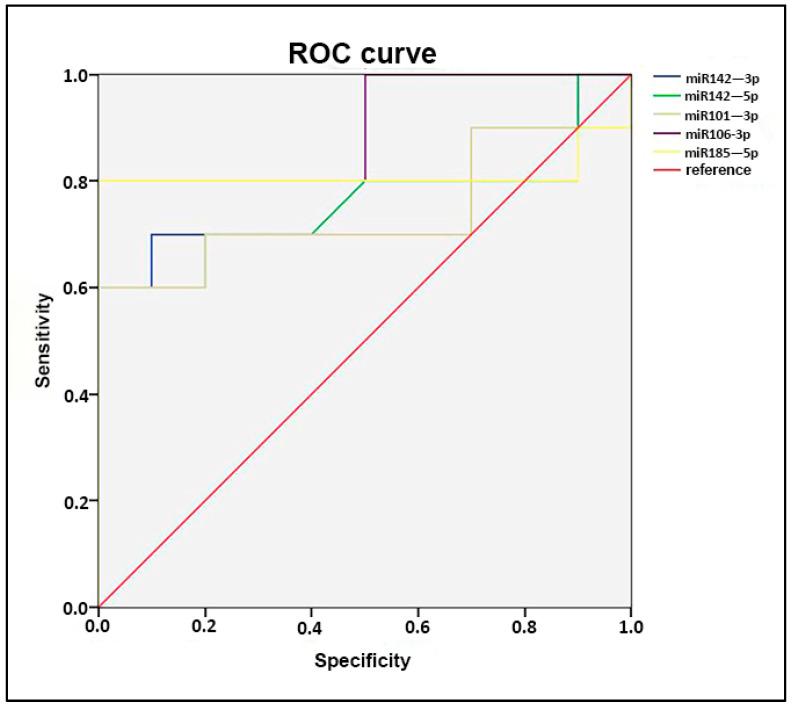
ROC curve. The ROC curve analysis highlighted the sensitivity of all 5 miRNAs and revealed that miR-106b-3p and miR-185-5p seem to be the ones better able to differentiate the two groups (SCR and no SCR). The oblique lines are derived from cases resulted to be equals.

**Figure 5 ijms-25-01911-f005:**
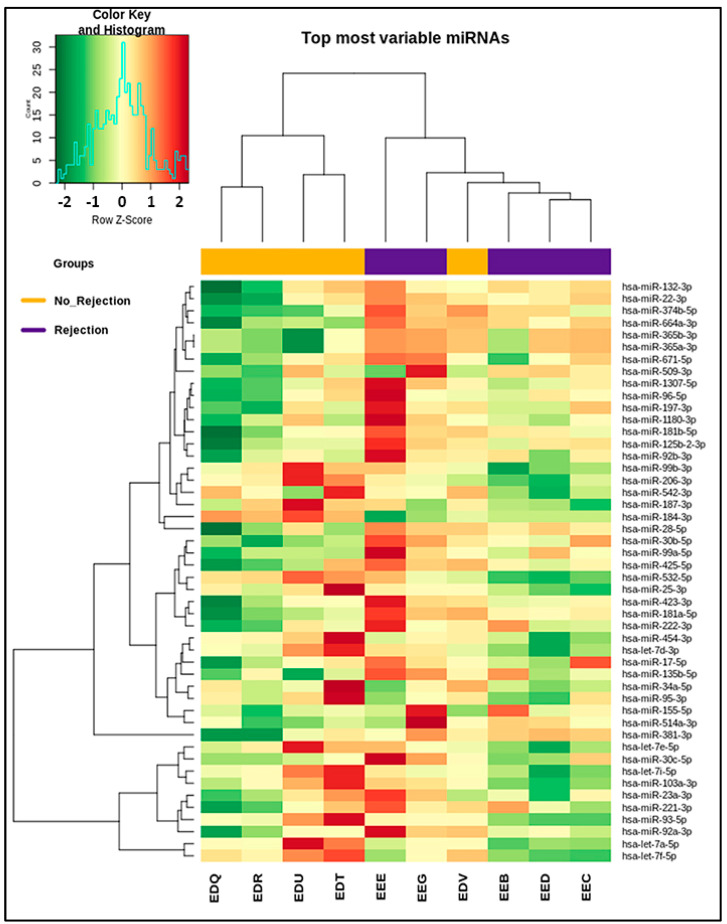
The heatmap represents the miRNAs differentially expressed in UEVs. The red color indicates upregulation, whereas green indicates downregulation. The two groups clustered perfectly, except for the EDV sample which shows a distinct trend, probably due to a drastically low expression compared to the rest of the cohort. (Patients with SCR: EEB, EEE, EEC, EED and EEG; patients with normal histology: EDR, EDT, EDU, EDQ and EDV).

**Table 1 ijms-25-01911-t001:** 5 miRNAs differentially expressed in tissue samples of the two subgroups (subclinical rejection vs. normal histology or no rejection) with *p*-value < 0.05.

miRNA	*p*-Value
hsa-miR-101-3p	0.0429
hsa-miR-185-5p	0.0428
hsa-miR-106b-3p	0.0315
hsa-miR-142-3p	0.0125
hsa-miR-142-5p	0.0059

**Table 2 ijms-25-01911-t002:** A total of 48 miRNAs differentially expressed among UEVs of two subgroups (subclinical rejection vs. normal histology or no rejection) with *p*-value < 0.05. miRNAs with positive logFC are upregulated in subclinical rejection (31 miRNA reported in bold), whereas negative logFC indicates upregulation in nonrejections (17 miRNA).

miRNA	logFC	*p*-Value
hsa-miR-184-3p	−1.641	0.000
**hsa-miR-99a-5p**	**1.112**	**0.000**
hsa-miR-93-5p	−0.709	0.000
hsa-let-7f-5p	−0.719	0.001
**hsa-miR-155-5p**	**1.325**	**0.001**
**hsa-miR-514a-3p**	**1.443**	**0.001**
hsa-miR-532-5p	−0.688	0.002
**hsa-miR-125b-2-3p**	**1.057**	**0.002**
**hsa-miR-1307-5p**	**0.612**	**0.003**
**hsa-miR-30b-5p**	**0.700**	**0.004**
**hsa-miR-22-3p**	**0.859**	**0.006**
hsa-let-7i-5p	−0.483	0.007
**hsa-miR-509-3p**	**1.165**	**0.007**
hsa-miR-187-3p	−0.914	0.008
**hsa-miR-423-3p**	**0.789**	**0.008**
**hsa-miR-132-3p**	**0.768**	**0.008**
**hsa-miR-181a-5p**	**0.886**	**0.009**
**hsa-miR-222-3p**	**0.792**	**0.010**
hsa-let-7a-5p	−0.558	0.010
**hsa-miR-664a-3p**	**0.944**	**0.011**
**hsa-miR-96-5p**	**0.572**	**0.011**
**hsa-miR-181b-5p**	**0.915**	**0.012**
hsa-miR-103a-3p	−0.374	0.015
**hsa-miR-92b-3p**	**0.894**	**0.016**
hsa-miR-542-3p	−0.819	0.017
**hsa-miR-30c-5p**	**0.487**	**0.018**
**hsa-miR-671-5p**	**0.628**	**0.022**
**hsa-miR-135b-5p**	**0.759**	**0.023**
hsa-miR-206-3p	−0.663	0.023
hsa-miR-99b-3p	−0.742	0.024
**hsa-miR-197-3p**	**0.625**	**0.025**
hsa-let-7d-3p	−0.445	0.025
**hsa-miR-23a-3p**	**0.402**	**0.025**
**hsa-miR-381-3p**	**1.103**	**0.026**
**hsa-miR-92a-3p**	**0.560**	**0.027**
hsa-let-7e-5p	−0.533	0.032
**hsa-miR-365a-3p**	**0.799**	**0.032**
**hsa-miR-365b-3p**	**0.799**	**0.032**
**hsa-miR-28-5p**	**0.883**	**0.032**
**hsa-miR-1180-3p**	**0.678**	**0.034**
**hsa-miR-374b-5p**	**0.735**	**0.035**
**hsa-miR-221-3p**	**0.629**	**0.037**
hsa-miR-34a-5p	−0.671	0.038
hsa-miR-95-3p	−0.734	0.041
**hsa-miR-425-5p**	**0.606**	**0.043**
hsa-miR-454-3p	−0.462	0.045
**hsa-miR-17-5p**	**0.486**	**0.049**
hsa-miR-25-3p	−0.450	0.049

## Data Availability

The data presented in this study are available on request from the corresponding author. The data are not publicly available due to privacy restrictions.

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
