# Peer review of "New Insights into Pediatric Kidney Transplant Rejection Biomarkers: Tissue, Plasma and Urine MicroRNAs Compared to Protocol Biopsy Histology"

_ijms, 2024, doi:10.3390/ijms25031911_

Round 1
Reviewer 1 Report
Comments and Suggestions for Authors
I carefully read the article by Carraro et al aimed at identifying miRNAs as markers of subclinical rejection after kidney transplantation in a paediatric population. Despite the problematic and consequent molecular biology work, the manuscript suffers from numerous flaws that do not allow us to have confidence in the results and conclusions provided. Extensive modifications are required in addition to minor changes relating to typos, syntax errors, or editorial nomenclature.
Regarding my most serious concerns:
- The study model for miRNAs associated with EVs is not justified by sufficient bibliographical sources. While EVs can protect miRNAs from degradation, many miRNAs are more present outside EVs than inside them, without being any less stable.
- We can also be concerned about the stability of the samples used. The patients were recruited between 2012 and 2015, but the study authorization dates back to 2022. How have the samples been stored in the meantime, and what is the impact of this long-term storage on the results?
- The study population and recruitment criteria are not sufficiently well described. The authors do not provide baseline characteristics, but assure us that the groups are homogeneous. This information may contradict the literature, which has shown that there are risk factors for subclinical rejection, and that homogeneity is not necessarily an expected outcome.
- Part 2.2 of the results: it is not clearly specified what these results correspond to: 1 patient? the average of all? only the SCRs? This needs to be made clearer, and the characteristics of individual samples and/or groups need to be provided.
- Figure 1: Western blots show the results of just 5 samples, without labels, showing that not all samples are of the same quality, or even contain the verification markers (CD63). The original WB images must be provided.
- Part 2.3 of the results: this part also lacks details that are not found in the methods either. What is the RIN? Where is "figure 24A"?
- Figure 2: the authors describe 5 patients with SCR whereas they announce 10 patients in this group line 129. Overall, having several cohorts and subgroup analyses does not make for easy understanding, and the presentation of results is not didactic enough.
- Figure 3 in Italian and not in English
- Lines 250-259: do not correspond to discussion points
- Many limitations of the study are not discussed.
- The study's design and methods do not support the article's conclusions, especially as the limitations are not explained/justified/discussed. I have ethical concerns about the inclusion of patients more than 10 years ago, a very recent ethics committee authorization, and the collection of consent (cf lines 327-328) provided by 10-year-old children.
- As mentioned above, sample collection and storage procedures are not specified, even though this may have an impact on results and conclusions.
- It is not relevant to have isolated serum and urine EVs according to different protocols and compared results. Especially since the ExoQuick-TC kit can be used to analyze urine. Moreover, the exoQUICK is supposed to isolate only exosomes and not all EVs. The section on EVS results should therefore be corrected accordingly. The results between urine and serum should not be compared and this methodological discrepancies must be taken in account when comparing to the litterature
Comments on the Quality of English LanguageFigure 3 in Italian and not in English
Author Response
Author's Reply to the Review Report (Reviewer 1)
Thank you for your accurate analysis of our study report.
To respond to your review, we preferred to proceed point by point (Review questions are in bold):
Firstlyt, a general revision of the manuscript was made for revise English and typos.
Regarding your most serious concerns:
- The study model for miRNAs associated with EVs is not justified by sufficient bibliographical sources. While EVs can protect miRNAs from degradation, many miRNAs are more present outside EVs than inside them, without being any less stable.
We have added two specifications from line 115 and a new reference: “Although miRNAs are considered good biomarkers, they could be unstable and easily degradable if they are free from ribonucleoprotein or lipoprotein particles. Therefore, the cellular miRNAs are often found included in extracellular vesicles (EVs) to avoid this degradation and ensure their function. The EV lipid membranes protect miRNA from degradation and remain stable in body fluids” (Yu, W et al. Exosome-Based Liquid Biopsies in Cancer: Opportunities and Challenges. Annals of Oncology 2021, 32, 466–477, doi:10.1016/j.annonc.2021.01.074.).
- We can also be concerned about the stability of the samples used. The patients were recruited between 2012 and 2015, but the study authorization dates back to 2022. How have the samples been stored in the meantime, and what is the impact of this long-term storage on the results?
As you understand, this is a retrospective study, for which notification was made to the ethics committee in 2021. Our center has banks of biological materials stored with the request for generic consent for conservation and research. These are precious samples, unique as they come from pediatric kidney transplant patients, collected during the protocol biopsy (a procedure that only 2 centers in Italy currently conduct for monitoring the transplanted organ). All tissue samples, present in this bank, collected between 2011 and 2016 were stored at -80°C with the use of an RNA stabilizer (RNALater®) which guarantees their long-term preservation. After extracting RNA samples from eligible patients, only those with intact RNA, identified through analysis with Agilent and RIN score, were selected for the study.
- The study population and recruitment criteria are not sufficiently well described. The authors do not provide baseline characteristics, but assure us that the groups are homogeneous. This information may contradict the literature, which has shown that there are risk factors for subclinical rejection, and that homogeneity is not necessarily an expected outcome.
The inclusion criteria were added to 4.1. Patients’ enrollment and study design paragraph.
- Part 2.2 of the results: it is not clearly specified what these results correspond to: 1 patient? the average of all? only the SCRs? This needs to be made clearer, and the characteristics of individual samples and/or groups need to be provided.
We improved 2.2. “EVS Extraction and Characterization” paragraph indicating that the EVS quantification data with NTA were ranges, and the diameters were the averages size of all samples. We have also added tables with individual values in the supplementary materials. We also improved the description of Figure 1.
- Figure 1: Western blots show the results of just 5 samples, without labels, showing that not all samples are of the same quality, or even contain the verification markers (CD63). The original WB images must be provided.
The original WB images were added to the supplementary materials.
- Part 2.3 of the results: this part also lacks details that are not found in the methods either. What is the RIN? Where is "figure 24A"?
RIN is RNA Integrity Number a quality parameter calculated by Agilent software. We added the description in 2.3. “RNA quality and concentration” paragraph. We also added 2 references concerning the RIN evaluation for RNA quality (Schroeder, A.; et al. The RIN: An RNA Integrity Number for Assigning Integrity Values to RNA Measurements. BMC Mol Biol 2006, 7, 3, doi:10.1186/1471-2199-7-3. And Helwa, I. et al. A Comparative Study of Serum Exosome Isolation Using Differential Ultracentrifugation and Three Commercial Reagents. PLoS One 2017, 12, e0170628, doi:10.1371/journal.pone.0170628.)
Sorry, Figure 24 does not exist, it was a typo error.
- Figure 2: the authors describe 5 patients with SCR whereas they announce 10 patients in this group line 129. Overall, having several cohorts and subgroup analyses does not make for easy understanding, and the presentation of results is not didactic enough.
We improved the description of Figure 2 and the description of the population and the 2 cohorts in 4.1. “Patients’ enrolment and study design paragraph. We have also included the Flowchart of the study design (Figure 5). Furthermore, to better understand the heatmaps, in the figures description we specified the acronyms related to patients with SCR and the one representing the patients with normal histology.
- Figure 3 in Italian and not in English
Sorry, we corrected it.
- Lines 250-259: do not correspond to discussion points
We moved the sentences to 4.1. “Patients’ enrollment and study design” paragraph.
- Many limitations of the study are not discussed.
We insert a limits discussion in 3. “Discussion” paragraph.
- The study's design and methods do not support the article's conclusions, especially as the limitations are not explained/justified/discussed. I have ethical concerns about the inclusion of patients more than 10 years ago, a very recent ethics committee authorization, and the collection of consent (cf lines 327-328) provided by 10-year-old children.
We hope that based on your suggestions too, the study design has been improved and we could make it clearer than before.
Regarding the samples used, as mentioned above, these are very valuable samples. For the tissues, we are confident that we have used a valid storage method, whereas, based on the results obtained on the sera, we think it is possible to improve their quality by adding for instance RNA Later for future studies. However, as specified previously, only patients who passed a quality check in the preparation of small RNA libraries were used.
Regarding the ethical concern, we must clarify that we did have a generic informed to store samples and for research purposes, as we mentioned above. We obtained more recently a notification enabling us to use these samples for this specific study.
Concerning the children’s authorizations, you are perfectly right, and we correct accordingly to it. Probably in revising the manuscript, some texts were missed, thus we added that the informed consent was signed by the parents.
- As mentioned above, sample collection and storage procedures are not specified, even though this may have an impact on results and conclusions.
Sample collection and storage have been described in 4.1. Patients’ enrollment and study design paragraph.
- It is not relevant to have isolated serum and urine EVs according to different protocols and compared results. Especially since the ExoQuick-TC kit can be used to analyze urine. Moreover, the exoQUICK is supposed to isolate only exosomes and not all EVs. The section on EVS results should therefore be corrected accordingly. The results between urine and serum should not be compared and this methodological discrepancies must be taken in account when comparing to the litterature
We apologize, there was a typo, we used ExoQuick-TM (not -TC) Exosome Precipitation Solution that can be used to purify exosomes from plasma, serum, and malignant ascites. We corrected the error in 4.2. “Serum EVs isolation paragraph.” We used this kit because of low input sample volume requirements. Of course, we agree with you, ExoQuick is a specific option for researchers who need to purify exosomes from small amounts of samples. Indeed, on purpose we declared to isolate EVs and not mention exosomes, as suggested by MISEV 2018; thus, we do not know their origins and we did not test their functions. Even the particle dimensions sustained what you underlined. Based on that to avoid confusion, we modified the text, specifying that these are small vesicles both in serum and urine results. Regarding the urine samples, we tried to use ExoQuick-TC, however, the output of EVS isolated was too low, so we decided to use an ultracentrifuge approach considering urine samples of 40 ml.

Reviewer 2 Report
Comments and Suggestions for Authors
I am writing to express my profound admiration for your groundbreaking research titled “New insight into pediatric kidney transplant rejection biomarkers: tissue, plasma, and urine microRNAs compared to protocol biopsy histology.”
Having delved into the intricacies of your study, I am compelled to acknowledge the meticulousness and significance of your work, particularly in its potential impact on enhancing the quality of care for pediatric kidney transplant recipients.
Your investigation into the early identification of subclinical rejection (SCR) fills a critical gap in the field of transplant medicine. The pursuit of alternative diagnostic methods, especially those that circumvent the invasiveness of protocol biopsies, is both commendable and necessary for improving patient outcomes. Your emphasis on the use of microRNAs (miRNAs), specifically in extracellular vesicles (EVs), presents a novel and promising avenue for non-invasive monitoring.
The identification of specific miRNAs, such as miR-142-3p, miR-142-5p, miR-101-3p, miR-185-5p, and miR-106b-3p, associated with histological SCR states in both tissue and urinary EVs, represents a significant leap forward. The correlation of these miRNAs with Banff histological classification adds a layer of reliability to your findings, showcasing the potential of miRNAs as valuable indicators of subclinical rejection.
One aspect of your research that warrants particular appreciation is the consideration of the pediatric population. The lack of a well-defined biomarker panel in this specific demographic makes your contribution all the more impactful. By investigating miRNA profiles in pediatric patients, your work addresses a crucial gap in existing literature and sets the stage for more tailored and effective interventions for this vulnerable group.
The potential of miRNAs to regulate immune responses and their implication in transplant rejection, as eloquently elucidated in your study, not only contributes to our understanding of the molecular mechanisms but also underscores their utility as non-invasive biomarkers. The protective role of extracellular vesicles in preserving miRNAs, coupled with their presence in different biological matrices, further reinforces the feasibility of utilizing miRNAs for diagnostic purposes.
Of particular interest is the potential application of your findings in improving the quality of life for pediatric kidney transplant patients. The prospect of a non-invasive diagnostic test for subclinical rejection, utilizing urinary EVs and avoiding the need for renal biopsies, holds immense promise. This could revolutionize clinical practice by providing a more patient-friendly and accurate means of monitoring transplant health.
As minor corrections I suggest to add the following two articles i the bibliography:
https://pubmed.ncbi.nlm.nih.gov/34563104/
https://pubmed.ncbi.nlm.nih.gov/34611792/
this addiction could improve the bibliography and corroborate your insights
Comments on the Quality of English Language
Minor editing
Author Response
Author's Reply to the Review Report (Reviewer 2)
Thank you for your accurate analysis and the appreciation expressed for our study.
In accordance to your suggestion, a general revision of the manuscript was made to correct both English and typos.
In addition, as you suggested, we improved the research design by explaining in detail the inclusion criteria for the patient's enrolment and a figure summarizing the study plan in the material and methods paragraph 4.1. Patients’ enrollment and study design paragraph.
To improve the presentation of the results we revised the 2.2. EVS Extraction and Characterization paragraph and we better define the Figure 1 description. We also added, as supplementary material: a) one table for sEVs and b) one for sUEVs indicating the EVs number and their obtained from NTA both; c) the original western blots for common markers for EVs.
We better describe the result of 2.3. RNA quality and concentration paragraph, adding also 2 references concerning the RIN evaluation for RNA quality (Schroeder, A.; et al. The RIN: An RNA Integrity Number for Assigning Integrity Values to RNA Measurements. BMC Mol Biol 2006, 7, 3, doi:10.1186/1471-2199-7-3. And Helwa, I. et al. A Comparative Study of Serum Exosome Isolation Using Differential Ultracentrifugation and Three Commercial Reagents. PLoS One 2017, 12, e0170628, doi:10.1371/journal.pone.0170628.)
Regarding the suggestion to improve the bibliography, we agreed with you, and we implemented it by adding two recent references about liquid biopsy in kidney transplants (“Seo, J.-W et al. Development and Validation of Urinary Exosomal MicroRNA Biomarkers for the Diagnosis of Acute Rejection in Kidney Transplant Recipients. Front Immunol 2023, 14, doi:10.3389/fimmu.2023.1190576.” and “Nassar, A.et al. Liquid Biopsy for Non-Invasive Monitoring of Patients with Kidney Transplants. Frontiers in Transplantation 2023, 2, doi:10.3389/frtra.2023.1148725.”).

Round 2
Reviewer 1 Report
Comments and Suggestions for Authors
Dear authors,
I carefully read your reply to my comments and the revised version of the manuscript.
Significant modifications were done on the manuscript to correct or discuss most of the limitations of the study
However, I cannot accept this for publication as I do not have access to supplementary materials. These data are mandatory to validate the study design and the subsequent results and conclusions
Author Response
Reviewer1: Dear authors,
I carefully read your reply to my comments and the revised version of the manuscript.
Significant modifications were done on the manuscript to correct or discuss most of the limitations of the study.
However, I cannot accept this for publication as I do not have access to supplementary materials. These data are mandatory to validate the study design and the subsequent results and conclusions.
Authors: Dear reviewer 1, we added the supplementary material as indicated by the journal. We are sorry you could not have access to them. Thus, here below (see the Word file) we report what we have added and submitted.

Reviewer 2 Report
Comments and Suggestions for Authors
You didn't improve the bibliography and corroborate your insights according to my suggestions.
You can improve it.
Comments on the Quality of English LanguageMinor editing.
Author Response
Thank you for your reply. The manuscript has been edited by a native English writer.
Reviewer 2: You didn't improve the bibliography and corroborate your insights according to my suggestions.
You can improve it.
Authors: Following what was suggested, in our previous response to the first revision of the article, we already improved the items "cited references relevant to the research", "research design", and "clarity in the presentation of the results" in the manuscript. However, you suggest again that to "improve the bibliography and corroborate our insights according to your suggestions".
We apologize for not being able to include the reference you mentioned, but these are the reasons.
Based on the links you attached, the two articles you mentioned were:
- (https://pubmed.ncbi.nlm.nih.gov/34563104/) Crocetto F, Cimmino A, Ferro M, Terracciano D. Circulating tumor cells in bladder cancer: a new horizon of liquid biopsy for precision medicine. J Basic Clin Physiol Pharmacol. 2021 Sep 27;33(5):525-527. doi: 10.1515/jbcpp-2021-0233.
- (https://pubmed.ncbi.nlm.nih.gov/34611792/) Sagnelli C, Sica A, Gallo M, Peluso G, Varlese F, D'Alessandro V, Ciccozzi M, Crocetto F, Garofalo C, Fiorelli A, Iannuzzo G, Reginelli A, Schonauer F, Santangelo M, Sagnelli E, Creta M, Calogero A. Renal involvement in COVID-19: focus on kidney transplant sector. 2021 Dec;49(6):1265-1275. doi: 10.1007/s15010-021-01706-6.
We carefully read them and although we found them very interesting, unfortunately, we could not find any relation or connection to our study. Indeed, the first study focused on bladder cancer and circulating tumor cells as a prognostic tool able to improve cancer patients’ clinical management. The second one is a review focusing on Sars-Cov2 infection in adult kidney transplanted patients.
Could it be that there was a wrong link and/or if you were referring to other studies?

Round 3
Reviewer 1 Report
Comments and Suggestions for Authors
Thanks to the authors for their prompt reply.
I noticed some points to discuss:
- Figure S1: there is an inconsistency between the type of EVs (sEVs) and the samples (EXX which correspond to UEVs). To be corrected
- How can you justify the analysis of samples 3 and 5 (Fig S1), which show no Evs markers and low concentrations according to Table S2?
Author Response
Thank you for your reply.
Reviewer1:
I noticed some points to discuss:
- Figure S1: there is an inconsistency between the type of EVs (sEVs) and the samples (EXX which correspond to UEVs). To be corrected
Authors: Thanks, we corrected in sUEVs
- How can you justify the analysis of samples 3 and 5 (Fig S1), which show no Evs markers and low concentrations according to Table S2?
Authors: We have included a note in the caption of Figure S1 of the supplementary materials, "Samples show a different markers expression. All samples were positive for at least one of the three markers. Flotillin-1 was the most present. The EDV sample shows a low marker expression, whereas the EEG sample seems to not have evidence of these proteins, probably due to its low concentration. This feature was commented on 2.2. Paragraph Extraction and Characterization of EVS." Furthermore, we added also a comment in 2.2. Paragraph Extraction and Characterization of EVS.
Supplementary materials (revised version)

Reviewer 2 Report
Comments and Suggestions for Authors
Authors answered comments and suggestions.
Comments on the Quality of English LanguageMinor editing
Author Response
Thank you for your reply.
Review2:
Comments on the Quality of English Language: Minor editing
Authors: A native English has re-edited the manuscript.
Round 4
Reviewer 1 Report
Comments and Suggestions for Authors
I read with attention the authors' last comments and corrections which are concordant with my suggestions